# Liquid foam improves potency and safety of gene therapy vectors

K. Fitzgerald [1], S. B. Stephan [1], N. Ma[2], Q. V. Wu [2] & M. T. Stephan [1,3,4] ✉

Interest in gene therapy medicines is intensifying as the first wave of gene-correcting drugs is now reaching patient populations. However, efficacy and safety concerns, laborious manufacturing protocols, and the high cost of the therapeutics are still significant barriers in gene therapy. Here we describe liquid foam as a vehicle for gene delivery. We demonstrate that embedding gene therapy vectors (nonviral or viral) in a methylcellulose/xanthan gum-based foam formulation substantially boosts gene transfection efficiencies in situ, compared to liquid-based gene delivery. We further establish that our gene therapy foam is nontoxic and retained at the intended target tissue, thus minimizing both systemic exposure and targeting of irrelevant cell types. The foam can be applied locally or injected to fill body cavities so the vector is uniformly dispersed over a large surface area. Our technology may provide a safe, facile and broadly applicable option in a variety of clinical settings.

Foam is becoming a prominent delivery system for small-molecule drugs to treat various medical conditions[1–3]. Due to their unique physicochemical properties, many foam products, such as Varithena®, Uceris®, or Luxiq®, have generated superior clinical outcomes and effectively replaced their liquid formulations[4–6]. Most recently, Arcutis Biotherapeutics scored FDA approval for a topical foam treatment, known as Zoryve®, for plaque psoriasis[7]. Despite their widespread use as carrier systems for pharmaceuticals and cosmetics, foams have not been studied as potential vehicles for therapeutic genetic material. This gap is quite astonishing, as foam possesses unique features (summarized in Fig. 1) that may yield superior potency and safety over the current delivery formulations for gene therapy. Foams are a special kind of colloidal dispersion in which closely packed gas bubbles are separated by thin layers of continuous liquid called lamellae. The large gas content and low fluid volume mean that an active ingredient will become highly concentrated in the lamellae (Fig. 1A). Thus, even with a small amount of therapeutic agent, large areas of tissue or body cavities can be exposed to high drug concentrations. This feature is especially relevant for gene therapy drugs, which are costly to manufacture and rely on high vector numbers at the target tissue to achieve efficient gene transfer and therapeutic benefit. The foam is stable and remains at the application site. This ensures a longer residence time for the gene therapy drug at the target tissue (Fig. 1B). Foams can also act as a local depot that slowly releases a gene therapy drug to produce enhanced tissue penetration (Fig. 1C).

Here, we explore these exceptional qualities of foam, with the goal of improving the effectiveness and safety of gene therapies while concomitantly lowering costs to help make these treatments widely accessible. We show that embedding nonviral- (mRNA-loaded lipid nanoparticles (LNPs)) or viral- (lentivirus) gene therapy vector within the liquid-filled lamellae of foam composed of methylcellulose and xanthan gum boosts transfection rates and permits precise payload delivery to defined locations. Using a clinically meaningful in vivo test system (intraperitoneal injection into immune-competent mice), we demonstrate that foam gene therapy, compared to liquid-based gene delivery, substantially improves in situ gene transfer and mediates a more homogeneous tissue transfection while limiting off-target exposure.

## Results

### Identifying a lead foam formulation

As a first step in developing this technology, we screened various compounds with known foaming properties for their ability to deliver

[1]Translational Science and Therapeutics Division, Fred Hutchinson Cancer Center, Seattle, WA 98109, USA. [2]Clinical Research Division, Fred Hutchinson Cancer Center, Seattle, WA 98109, USA. [3]Department of Bioengineering and Molecular Engineering & Sciences Institute, University of Washington, Seattle, WA 98109, USA. [4]Department of Medicine, Division of Medical Oncology, University of Washington, Seattle, WA 98195, USA. ✉ e-mail: mstephan@fredhutch.org

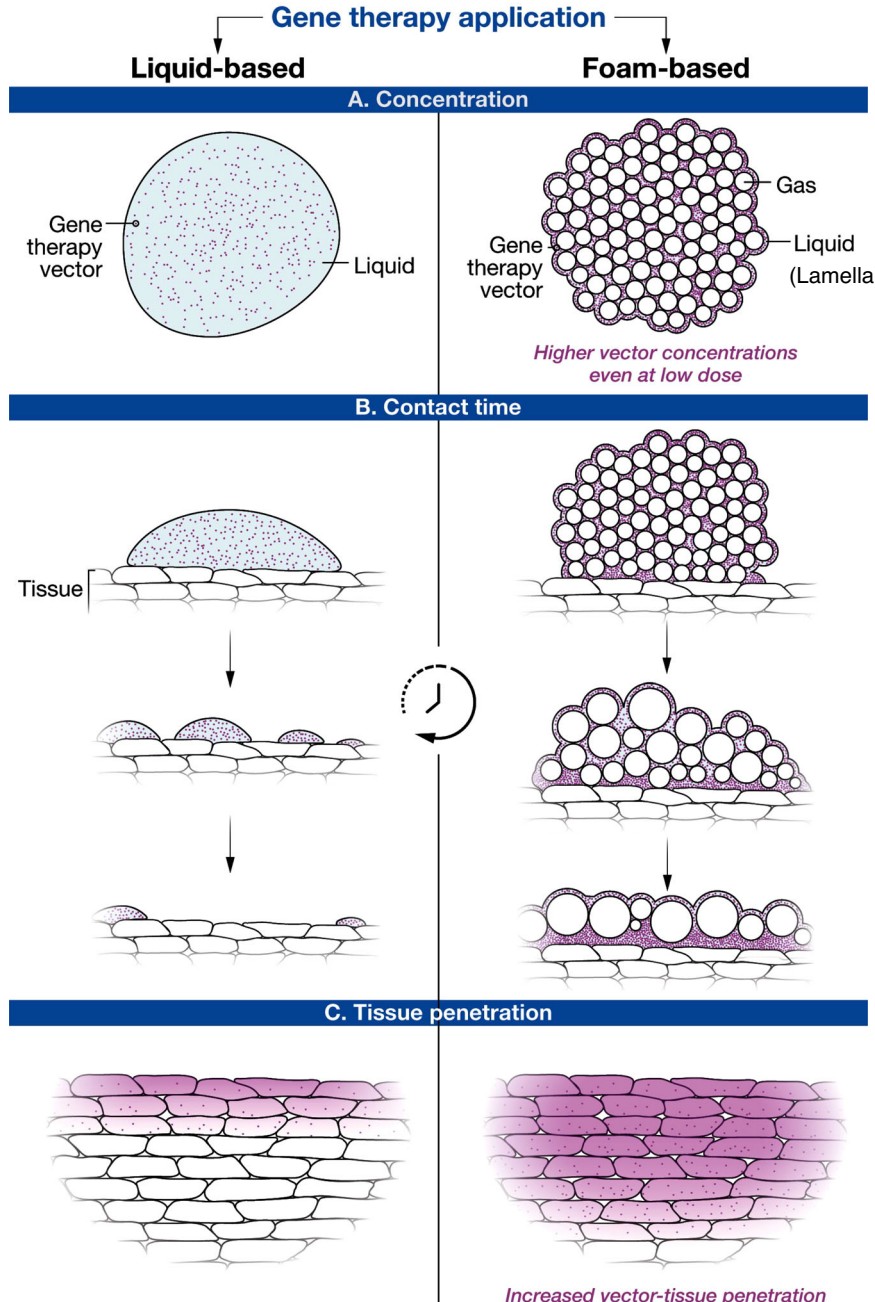

**Fig. 1 | Schematic explaining the key advantages of foam as a gene delivery system in comparison to conventional liquid formulations. A** Foam is mostly gas, so the embedded vector particles become heavily concentrated in its liquid component, which ensures high-density exposure of target tissue to the gene therapy vector. **B** Foam remains at the application site longer, thereby enhancing the delivery of the gene therapy drug to the intended cells and minimizing unwanted off-target effects. **C** Higher vector density combined with longer contact time results in higher transfection rates and deeper tissue penetration.

gene therapy vectors to human cells. We only considered candidates that are generally regarded as safe by the Food and Drug Administration and are already manufactured in bulk, mostly for the food industry but also the pharmaceutical industry. This ensures that the foam vehicle we are formulating is biocompatible, scalable, and low-cost. One plant-based food additive we tested is methylcellulose, which is a widely used foaming agent in foods like pudding, ice cream, and whipped toppings[8]. Our second candidate, sodium caseinate, the sodium salt of casein (a milk protein), is used in foods and cosmetics for its foaming and thickening properties[9]. We chose human serum albumin as the starting material of our third foam formulation. This globular protein is routinely administered to patients with acute

conditions such as trauma, cardiogenic shock, and sepsis, but it is also being explored by bioengineers for its foaming properties[10,11]. To make our gene therapy foam more stable for therapeutic purposes, we added xanthan gum to all three tested foam precursor solutions. Xanthan gum is an FDA-approved polysaccharide used as a foam stabilizer in foods, cosmetics, and healthcare products[12–14].

Our initial experimental protocol for in vitro transfection screening involved delivering a nonviral vector. We fabricated LNPs based on the formulation of Moderna's COVID-19 mRNA vaccine, replacing the mRNA encoding COVID antigen with Luciferase mRNA (Supplementary Fig. 1)[15]. This allows for easy comparison of transfection efficiencies using bioluminescence after adding these LNPs either

suspended in liquid or embedded in various foam formulations to cultured cells. Cells were incubated with LNP suspensions or LNP foam for two hours, during which the culture dish was placed in a horizontal or tilted position. The horizontal setup is designed to compare gene transfer efficiencies in the absence of any liquid drainage, whereas the angled transfection mimics the more realistic scenario of patient tissue that is shifting in position and lacks defined borders that would prevent drainage of the applied therapeutics (Supplementary Fig. 2). In the horizontal transfections, we found that embedding mRNA LNPs in foam made of methylcellulose boosted their transfection efficiency by 2.9-fold ($P < 0.0001$) compared to LNP suspension (Fig. 2a, b). This effect was even more pronounced in the angled cell transfection, where this foam formulation achieved a robust 384-fold higher gene transfer (Fig. 2a, c). Importantly, in all our in vitro transfection studies, the viability of cells following extended contact with gene therapy foam was unaffected (Fig. 2d). Other tested foams did not improve the transfection potential of mRNA LNPs at the levels we observed with methylcellulose (Fig. 2a–c). Furthermore, because methylcellulose foam stays in place, the transfections were spatially well-defined. This is vividly illustrated by our ability to inscribe text onto cells grown in a tissue culture plate while holding the dish vertically, which then appeared as an identical pattern of gene expression 24 h later (Fig. 2e). Based on all these in vitro results, we selected the methylcellulose foam formulation as our lead candidate for further analysis and studies.

## Foam characterization

To precisely measure the foam structure in terms of bubble size, bubble distribution, and foam decay over time, we employed a Dynamic Foam Analyzer DFA100FSM (Krűss Scientific). This instrument uses contact-free optical sensors and an LED light source to continuously record the height of the foam, liquid drainage, and the sizes and distributions of foam bubbles to a precision of 0.1 mm. Based on three independently manufactured foam batches, we measured an initial mean bubble count of $95.3 \pm 4.2/mm^2$ and a mean bubble area of $10,528 \pm 455\ \mu m^2$ (Fig. 3a). As expected, the larger bubbles grew with time at the expense of smaller bubbles due to diffusion of air from the smaller to the larger, resulting in a decrease in bubble count ($25.7 \pm 6.7/mm^2$) and an increase in bubble size (mean $41,597 \pm 9,387\ \mu m^2$) after 10 h at room temperature (Fig. 3a). Foam height (total height – liquid height) decayed to an average $78 \pm 9\%$ of the maximum value. LNPs embedded in the foam were relatively homogenously dispersed throughout the liquid lamellae, as visualized by confocal microscopy (Fig. 3b, c). Based on 27 analyzed foam lamellae, we measured a slight increase (1.1-fold, $P < 0.0001$) in LNP concentration in centers compared to edges adjacent to the gas bubbles (Fig. 3d).

## Benefit of foam for nonviral vector administered intraperitoneally

Based on these in vitro studies, we next evaluated the benefit of foam as a gene delivery vehicle in relevant in vivo test systems. We tested the intraperitoneal route of administration, which could be used to treat a variety of pathologies or organs in the peritoneal cavity, including cancers such as ovarian, pancreatic, liver, colorectal, and gastric. Since the abdominal cavity is a closed anatomic space, the options for LNP drainage are limited to the abdominal lymphatics. This means that whether delivered by liquid or foam, the residence time of the gene therapy vector is similar. Therefore, the question we wanted to address in these in vivo studies is whether a foam formulation intrinsically achieves higher in situ transfection rates than a liquid suspension, i.e., independent of the foam's ability to stay in place at the application site. Mice were divided into groups that received an equal dose of LNPs containing luciferase-encoding mRNA administered in PBS solution or in foam, both freshly prepared just prior to injection. We then used bioluminescence imaging to quantify gene expression over a four-day

period. We found that foam injections resulted in an overall average 3.2-fold higher transfection rate compared to liquid-based gene therapy (Fig. 4a, b). As expected with mRNA-based vectors, luciferase was transiently expressed, showing the highest level of expression 24 h post-injection. Direct visualization of gene transfer into various organs collected from the peritoneal cavity confirmed substantially higher transfection rates achieved with foam versus liquid (Fig. 5a, b). Flow cytometric analysis of peritoneal and splenic macrophages also revealed higher transfection rates (2.94-fold and 3-fold increase, respectively) in foam-injected mice. (Supplementary Fig. 3). Notably, the use of foam resulted not only in highly efficient but also homogeneous gene transfer, which was most obvious in the small intestine, as it displayed consistent transgene expression along its entire length in all injected animals (Fig. 5a) as a result of homogenous LNP distribution (Fig. 5c, d). In contrast, LNP suspensions produced low and irregular gene transfer, with most sections of the intestine untransfected, reflecting the absence or low density of LNPs in those sections (Fig. 5c, d).

## Biocompatibility

Guided by these imaging data, we next conducted a comprehensive toxicity assessment. Mice were injected intraperitoneally with foam or PBS and euthanized 48 h later to collect blood for serum chemistry and to perform a complete gross necropsy. The following tissues were evaluated by a board-certified staff pathologist: intestine, liver, spleen, mesentery, pancreas, and stomach. Serum chemistry was unremarkable (Fig. 6a). The few histological lesions noted in foam-treated animals were minimal to mild. In four of the ten mice injected with foam, sections of the mesenteric fat were infiltrated by a mixture of neutrophils and histiocytes (Fig. 6b). We also noticed in all foam-treated mice an expansion of the marginal zone and red pulp of the spleen (Fig. 6c). The pathologist who analyzed the samples explained to us that the marginal sinus ectasia is presumed to be a drainage response to the viscous liquid foam. Overall, these data establish that our methylcellulose-based foam does not trigger clinically significant local or systemic toxicities and is a safe delivery vehicle for gene therapy vectors.

## Foam as a carrier of viral gene therapy vector

To confirm that this technology also has relevance to gene therapy using viral vectors, we next investigated the benefit of foam as a delivery vehicle for Lentivirus (LV). LVs have proven their safety and efficacy as flexible gene delivery vectors in clinical applications of gene therapy for genetic diseases[16]. Traditionally, LV vectors have been used to transduce patient-derived somatic cells ex vivo[16], which are then re-delivered to the same patient. However, a rapidly growing number of trials are currently investigating LVs for in vivo gene therapy to treat monogenic diseases and chronic diseases, including neurological[17], ophthalmological[18], and metabolic disorders[19]. In our studies, we employed a commonly used, broadly tropic vesicular stomatitis virus envelope glycoprotein (VSV-G) pseudotyped LV vector with the luciferase transgene driven by the Cytomegalovirus (CMV) promoter. An equal dose of viral particles ($4 \times 10^6$ transduction units) either suspended in PBS or embedded in methylcellulose foam was injected into the peritoneal cavity of mice, and gene expression was serially quantified by bioluminescence imaging. Two days after vector administration, we found that the foam improved LV-mediated in situ gene transfer by an average 10.2-fold (Fig. 7a, b). Gene expression levels were tracked for four weeks, during which time mice inoculated with LV-foam maintained similarly high differences in luciferase levels over mice treated with LV suspended in PBS (average 9.8-fold increase over LV-PBS during weeks 1-4, Fig. 7b). Quantification of viral particles in the peritoneal cavity versus the peripheral blood by ELISA three hours post injection revealed that foam retained an 89.4-fold higher concentration of LV gene therapy vector at the application site, compared to liquid suspension (Fig. 7c, left panel). Conversely, we measured a 2.9-

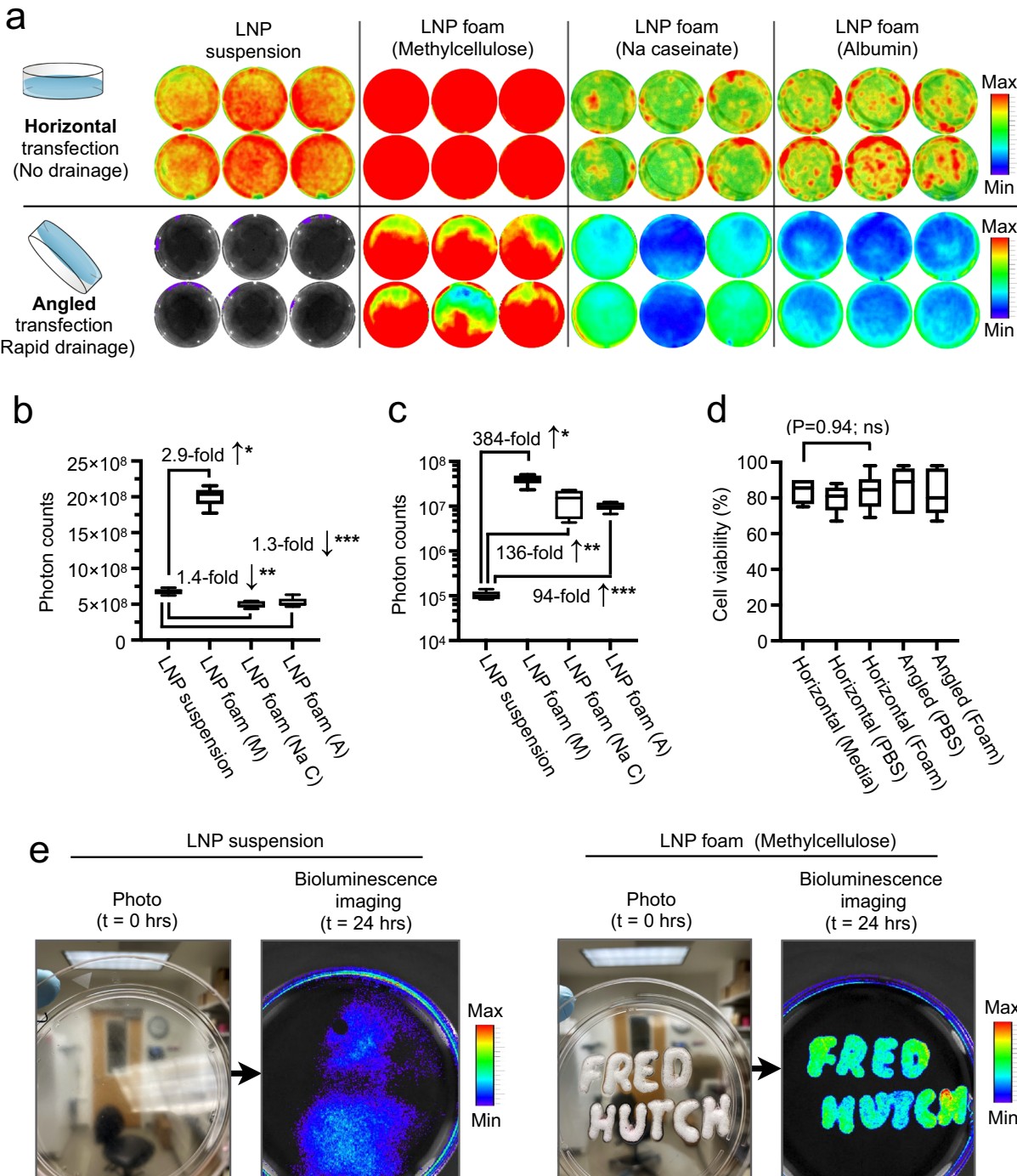

**Fig. 2 | Gene therapy foam boosts transfection rates and permits precise payload delivery to defined locations. a–d** Screening of various foam vehicles to enhance topical gene delivery. Lipid nanoparticles (LNPs) carrying firefly luciferase-encoding mRNA were mixed with foam precursor solutions composed of 0.8 wt % methylcellulose, sodium caseinate (Na caseinate), or albumin. Prepared foam was added on top of cultured HeLa cells, or cells were transfected with an equal dose of LNPs suspended in PBS. **a** In vitro bioluminescent imaging of cells exposed to LNP suspension or LNP foam. Boxplots summarizing luciferase signals from horizontal (**b**) and angled (**c**) transfections. On each box plot, the central mark indicates the median, and the bottom and top edges of the box indicate the interquartile range. Whiskers represent 95% confidence intervals. **b** *$P < 0.0001$; **$P < 0.0001$; ***$P = 0.0004$. **c** *$P < 0.0001$; **$P = 0.0014$; ***$P < 0.0001$. Images of cell culture dishes shown on the left of **a** were created with BioRender.com. (**d**) Quantification

of viable (trypan blue-excluding) HeLa cells following horizonal or angled transfections with PBS or methylcellulose foam. On each box plot, the central mark indicates the median, and the bottom and top edges of the box indicate the interquartile range. Whiskers represent 95% confidence intervals. Pairwise differences between the groups were statistically analyzed using the unpaired, two-tailed Student's $t$ test. $P < 0.05$ was considered significant. Ns., non-significant. $N = 6$ biologically independent samples. **e** Photos and corresponding bioluminescence images of cell culture dishes seeded with HeLa cells. The text "FRED HUTCH" was written onto cells using syringes loaded either with LNP suspension or LNP foam, while holding the culture plates vertically. Bioluminescent signals were visualized 24 h later. The shown pictures represent four dishes each with similar outcomes. Source data are provided as a Source Data file.

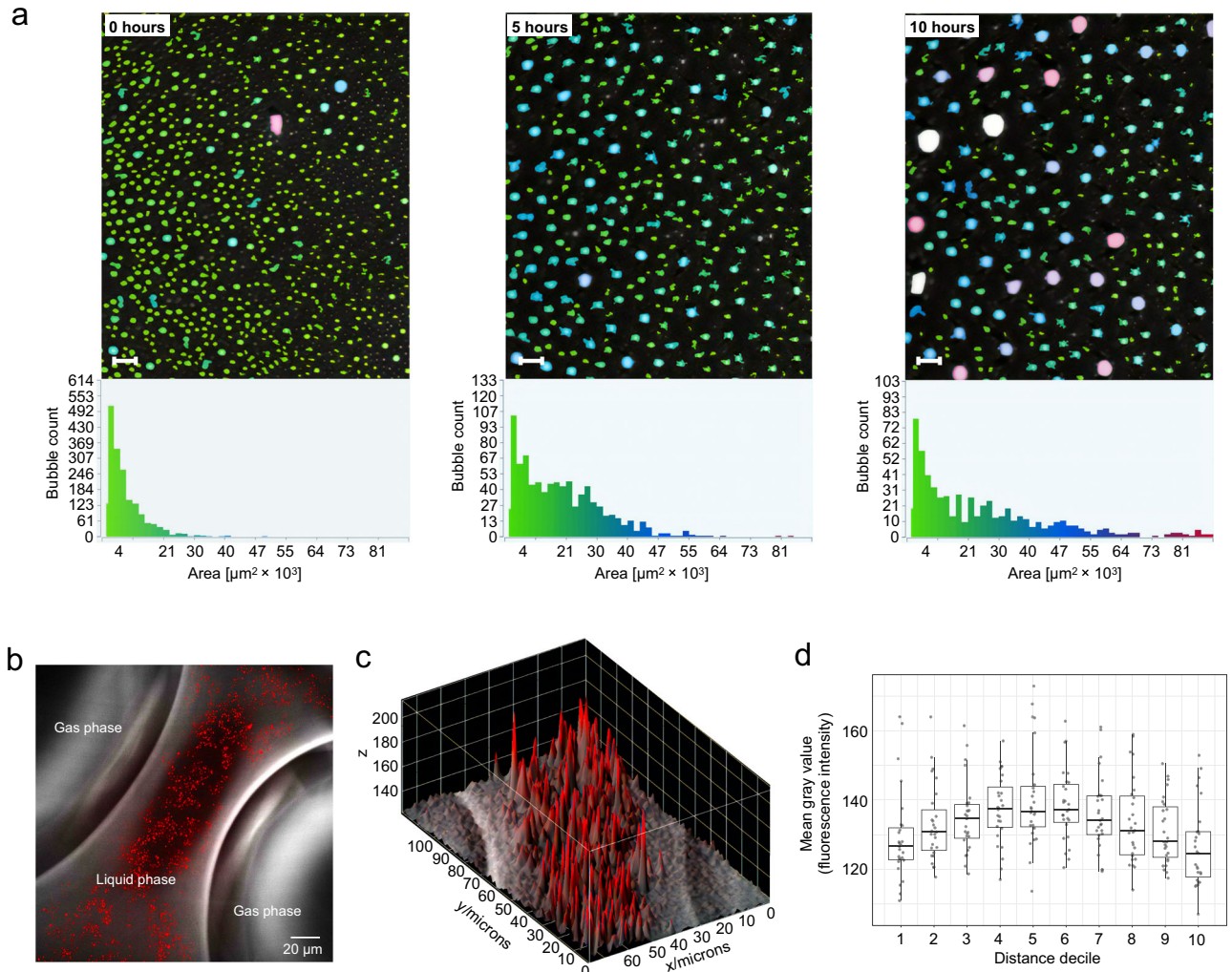

**Fig. 3 | Microstructural characterization of LNP-loaded methylcellulose foam.** **a** High-resolution foam structure analysis using a Dynamic Foam Analyzer DFA100FSM. Representative images of three independently manufactured foam batches are shown at time points 0 h, 5 h, and 10 h. Histograms of bubble size distribution are shown below. Scale bars: 1 mm. **b** Confocal microscopy image of a liquid-filled foam lamella separating two gaseous bubbles. To visualize the distribution of LNPs within the lamella, we tagged their lipid envelope with fluorescent 1,1-dioctadecyl-3,3,3,3-tetramethylindodicarbocyanine (DiD'). The shown image represents 50 independent acquisitions. **c** DiD' intensities in the foam lamella were analyzed and their spatial distribution plotted in 3D. Data are representative of 27 independently characterized lamellae. **d** For each image, we divided the distance between two air bubbles into ten equal segments and calculated the average particle density within each segment. The boxplots showing the mean fluorescence intensities of each segment for 27 independent images (distance decile 1 refers to the first segment from the left side and distance decile 10 refers to the 10th segment from the left, which is also the first segment from the right side). Boxes extend from the 25th to the 75th percentile of each group's distribution of values. The thick black line within each box denotes the median value. Whiskers mark adjacent values within 1.5 interquartile range (IQR) from 25th and 75th percentile. Source data are provided as a Source Data file.

fold higher off-target viral count in the peripheral blood of animals treated with LV suspension (Fig. 7c, right panel). These data establish that foam can mitigate unwanted systemic gene therapy drug exposure and focus its action on therapeutically desired tissue.

## Discussion

To the best of our knowledge, no other scientific report has considered foam for gene therapy. Preclinical studies have explored foam for the release of chemotherapy or oxygen within tumors[20,21]. Also, a carbon monoxide-releasing therapeutic foam has recently been described[22,23]. While gene therapy foam is clearly not suited for systemic infusion, the potential clinical applications of this foam platform are numerous and include improving the safety and potency of oncolytic virus therapy, enhancing vaccines, developing in situ gene therapy for gastrointestinal diseases (oral cancer, esophageal cancer, stomach cancer, colorectal cancer, autoimmune diseases that affect the digestive

system), gynecological cancer, skin disease (in particular wound healing), mesotheliomas, cancers spreading to the peritoneal cavity, or any kind of in situ gene modification that requires topical application.

Figure 8 depicts the envisioned clinical workflow for generating and applying the gene therapy foam. A syringe filled with foam precursor solution and gene therapy vector is connected to a syringe containing air (Fig. 8A). The foaming liquid and gas are then moved repeatedly back and forth through the constriction that connects the two syringes, generating a homogenous foam which contains evenly distributed therapeutic vector. This medication is then directly administered to the patient to form local reservoirs that slowly release the gene therapy drug (Fig. 8B). This workflow is similar to that for Varithena®, an FDA-approved drug/delivery unit that produces foam containing 1% polidocanol for the management of varicose veins[4]. Foam sclerosants like Varithena® have effectively replaced liquid agents because they produce better clinical outcomes[24]. Medical

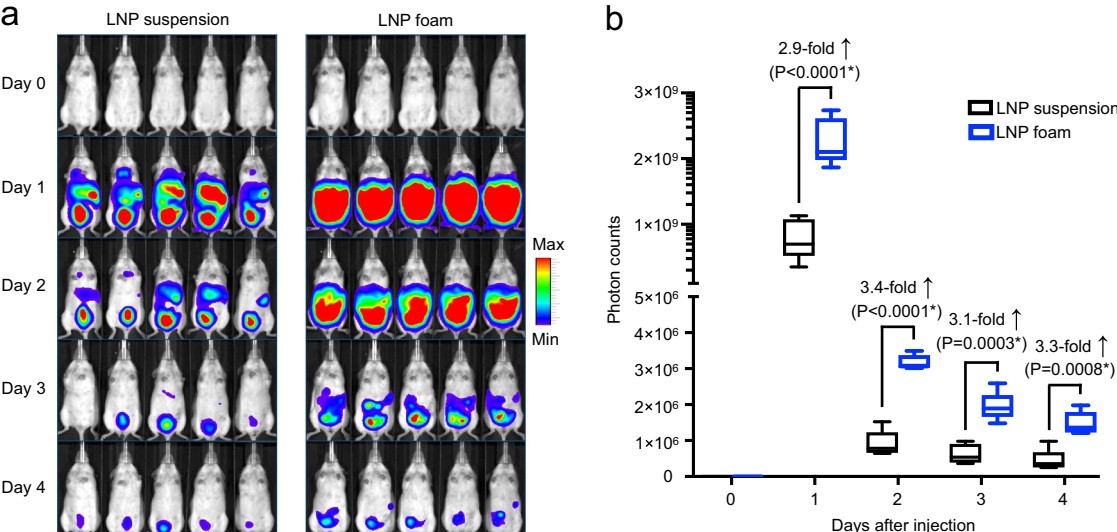

**Fig. 4 | Foam as a carrier system for gene therapy vectors can improve in situ gene transfer.** Immunocompetent C57BL/6 albino mice were injected intraperitoneally with a single dose of LNPs suspended in PBS or incorporated into methylcellulose foam. To noninvasively track gene expression in vivo, LNPs were loaded with luciferase-encoding mRNA. **a** Sequential bioluminescence imaging of gene expression. **b** Boxplots showing photon counts from luciferase activity. On each box plot, the central mark indicates the median, and the bottom and top edges of the box indicate the interquartile range. Whiskers represent 95% confidence intervals. $N = 5$ biologically independent samples. Pairwise differences between the groups were statistically analyzed using the unpaired, two-tailed Student's $t$ test. $P < 0.05$ was considered significant (*). Ns., non-significant. Source data are provided as a Source Data file.

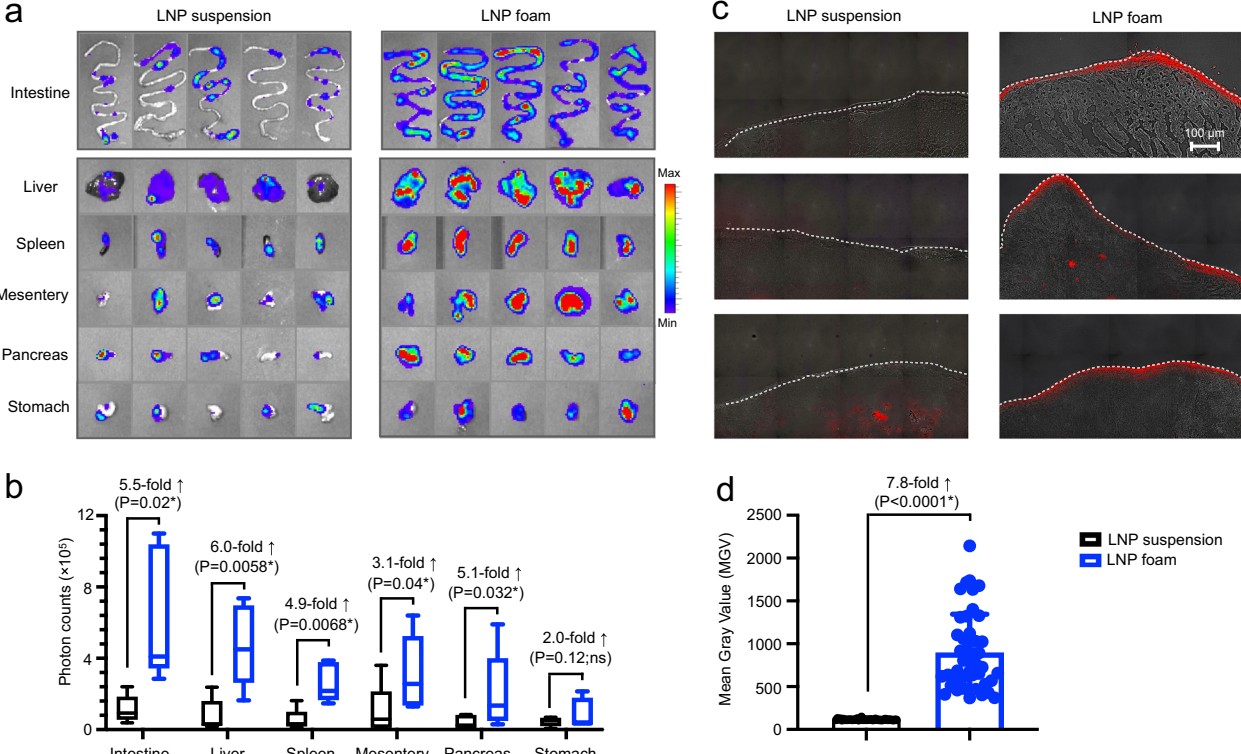

**Fig. 5 | Foam mediates homogeneous tissue transfection. a** IVIS images of organs and tissues isolated from mice 24 h following intraperitoneal administration of LNPs containing Luciferase mRNA. Injected LNPs were either suspended in PBS or incorporated into freshly prepared methylcellulose foam. Luciferase signals from each organ were quantitated and graphed in **b**. On each box plot, the central mark indicates the median, and the bottom and top edges of the box indicate the interquartile range. Whiskers represent 95% confidence intervals. $N = 5$ biologically independent samples. Pairwise differences between the groups were statistically analyzed using the unpaired, two-tailed Student's $t$ test. $P < 0.05$ was considered significant (*). Ns., non-significant. **c** TissueFAXS imaging of mouse intestines 24 h following intraperitoneal administration of LNPs fluorescently labeled with DiD' (shown in red). Injected LNPs were either suspended in PBS or incorporated into freshly prepared methylcellulose foam. DiD' signal intensities (Mean Gray Value, which is provided by the Measure tool in ImageJ and is the sum of the gray values of all the pixels in the selection divided by the number of pixels) are plotted in **d**. Error bars represent standard deviation of the mean. $N = 44$ independently analyzed sections per group. Pairwise differences between the groups were statistically analyzed using the unpaired, two-tailed Student's $t$ test. $P < 0.05$ was considered significant (*). Source data are provided as a Source Data file.

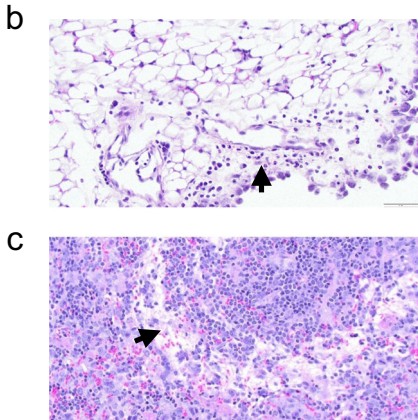

**a**

| Parameter | Unit | PBS control | Foam |
|---|---|---|---|
| PCV | % | 43.8±6.6 | 46.6±3.2 |
| WBC | K/μL | 4.2±0.6 | 5.2±2.0 |
| NEUCT | /μL | 460±95 | 667±183 |
| LYMCT | K/μL | 3.6±0.6 | 3.8±1.0 |
| MONCT | /μL | 180±50 | 317±102 |
| BASCT | /μL | 43±36 | 37±22 |
| EOSCT | /μL | 70±46 | 60±39 |
| ALP | U/L | 113±40 | 41±6.4 |
| ALT | U/L | 38±17 | 63±55 |
| AST | U/L | 212±106 | 377±281 |
| BUN | mg/dL | 21±5 | 15±3 |
| CREA | mg/dL | 0.4±0.1 | 0.3±0.1 |
| GLU | mg/dL | 204±46 | 166±45 |

**Fig. 6 | Methylcellulose-based foam is biocompatible.** These panels summarize a pathology report prepared by a Comparative Pathologist at Fred Hutch. Two days after a bolus injection of methylcellulose foam or PBS into the peritoneal cavity of mice, various key organs close to the peritoneum as well as blood were isolated for a double-blinded analysis. **a** Serum chemistry and blood counts. Data are mean ± s.d.; $N = 5$ biologically independent samples. **b** Representative hematoxylin and eosin (H&E)-stained sections of the mesenteric fat from mice that exhibited mild regional infiltrates of neutrophils and histiocytes in response to foam injection. **c** Representative image of the marginal zone of a spleen 48 h after intraperitoneal foam injection. ×40 objective. $N = 10$ biologically independent samples. Source data are provided as a Source Data file.

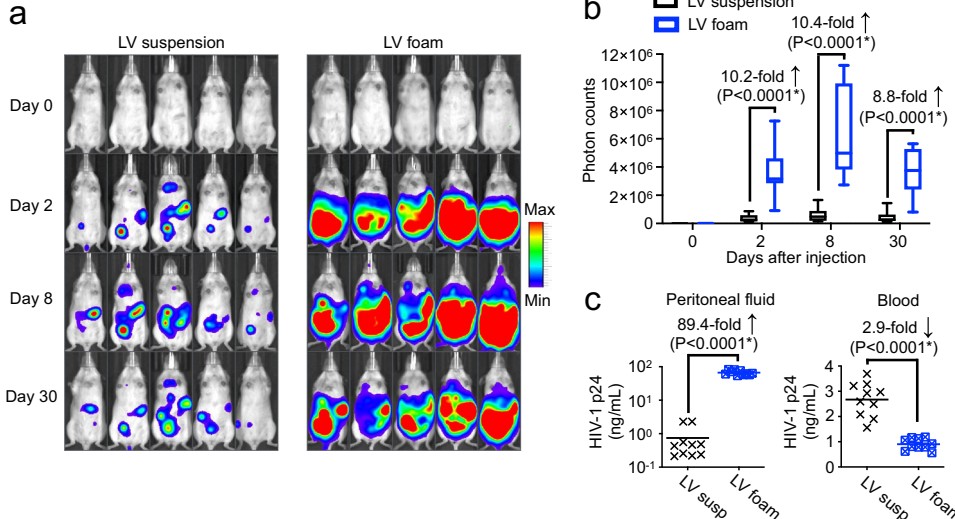

**Fig. 7 | Foam amplifies lentivirus-mediated gene transfer while limiting undesirable systemic exposure.** C57BL/6 albino mice were injected intraperitoneally with a single dose of $5.25 \times 0^6$ lentiviral particles (LV) suspended in PBS or incorporated into methylcellulose foam. To noninvasively track gene expression in vivo, LV expressed luciferase. **a** Representative sequential bioluminescence imaging of gene expression. **b** Boxplots showing photon counts from luciferase activity. On each box plot, the central mark indicates the median, and the bottom and top edges of the box indicate the interquartile range. Whiskers represent 95% confidence intervals. $N = 10$ biologically independent samples. Pairwise differences between the groups were statistically analyzed using the unpaired, two-tailed Student's $t$ test. $P < 0.05$ was considered significant (*). **c** Lentivirus vector concentrations in the peritoneal fluid and peripheral blood 3 h post injections. Error bars represent standard deviation of the mean. $N = 10$ biologically independent samples. Pairwise differences between the groups were statistically analyzed using the unpaired, two-tailed Student's $t$ test. $P < 0.05$ was considered significant (*). Source data are provided as a Source Data file.

practitioners routinely prepare sclerosant foam at the bedside by agitating liquid sclerosant with gas. Thus, we anticipate our gene therapy strategy could be swiftly adopted into clinical practice to address critical gaps that limit the application of gene therapy:

1. **The problem of practicality:** As far as we know, there are currently no methods in the clinic that allow the delivery of gene therapy vectors to specific target cells in tissue. FDA-approved in situ gene therapies either expose the patient to vector systemically (e.g., Elevidys® for Duchenne Muscular Dystrophy[25]) or inject vector directly into a confined space, such as the bladder (Adstiladrin®[26]), the spinal canal (Qalsody® for amyotrophic lateral sclerosis[27]) or the eyeball (Luxturna® for retinal disease[28]). In the latter three treatments, the vector has little opportunity to leak and thus is able to unload its DNA cargo into a maximum number of target cells. However, this treatment approach is not practical for most organs, which are co-located within large body cavities (e.g. abdominal, thoracic), or are tissues that line the gastrointestinal or reproductive tract, where vector would not be retained at the application site. Our foam approach provides controlled release of gene therapy agents at desired locations. Foam can be applied locally, or injected to fill body cavities so the vector is uniformly dispersed over a large surface area.

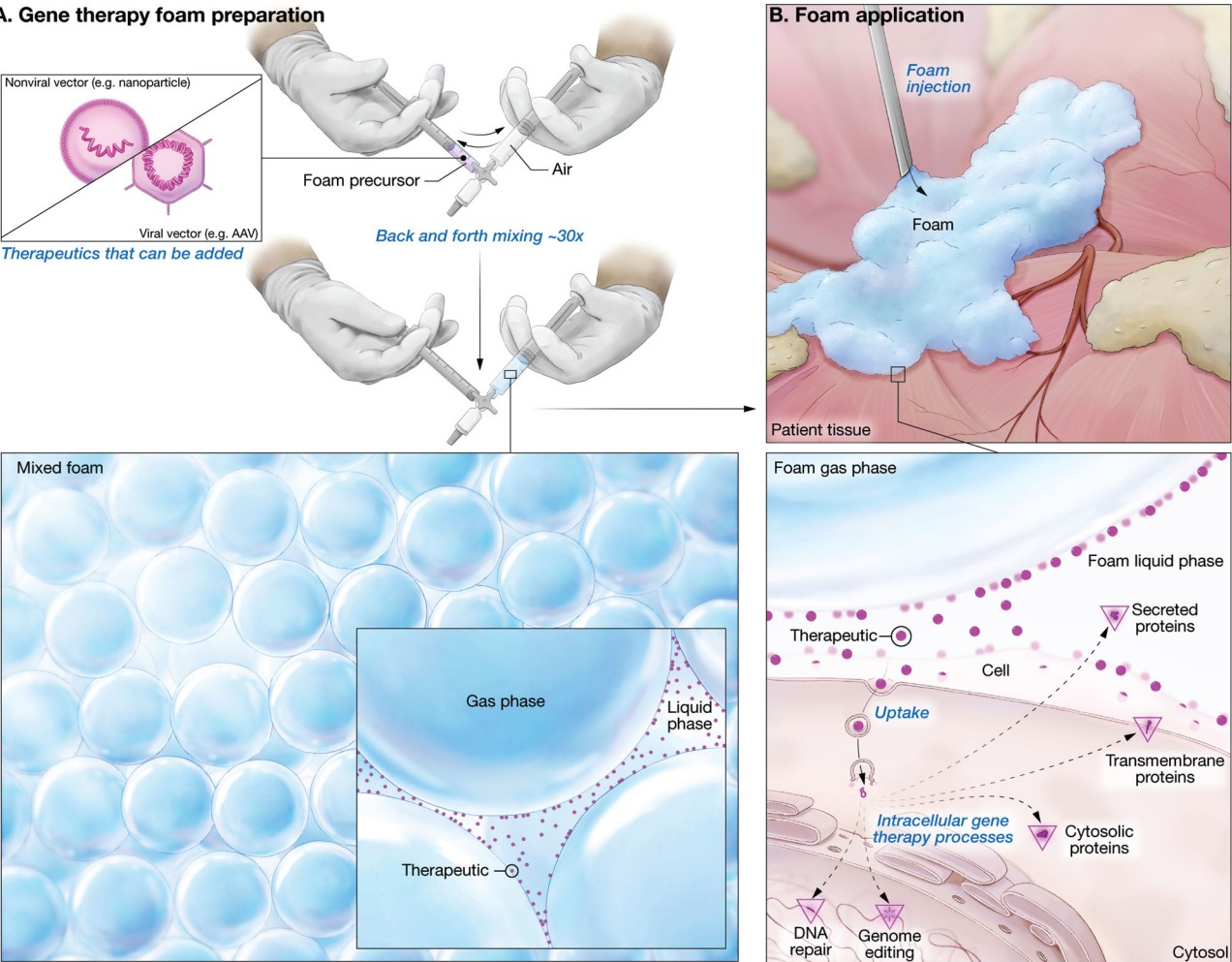

**A. Gene therapy foam preparation**

Nonviral vector (e.g. nanoparticle)

Viral vector (e.g. AAV)

*Therapeutics that can be added*

Foam precursor

Air

*Back and forth mixing ~30x*

Mixed foam

Gas phase

Liquid phase

Therapeutic

**B. Foam application**

*Foam injection*

Foam

Patient tissue

Foam gas phase

Foam liquid phase

Therapeutic

*Uptake*

Cell

Secreted proteins

Transmembrane proteins

*Intracellular gene therapy processes*

Cytosolic proteins

DNA repair

Genome editing

Cytosol

**Fig. 8 | Schematic illustration depicting how gene therapy foam is freshly prepared and applied therapeutically to supply new genetic material or change existing DNA in cells. A** Nonviral or viral vector (Therapeutic) is added to foam precursor in a syringe connected to a second syringe filled with air. The air and foam precursor are mixed by vigorously drawing the syringe plungers back and forth at least 30 times, creating a uniform microfoam consisting of gas bubbles separated by a network of interconnected liquid film structures called lamellae. Gene therapy vectors are concentrated in this liquid phase as the foam matures. **B** Once applied to tissue, the foam gradually deploys its therapeutic cargo and either supplies new genetic material or changes the endogenous DNA in the target cell.

2. **The problem of cost and accessibility:** Gene therapies are the most expensive drugs in the world. The retinal therapy Luxturna® was launched at a cost of $425,000 per eye, for which the patient receives $1.5 \times 10^{11}$ vector genomes of an adeno-associated virus (AAV) resuspended in $0.3\,\mathrm{mL}$[28]. Most other clinical applications will require much larger volumes and vector doses, likely commanding even higher prices. Therefore, many gene therapies will be cost-prohibitive or available to only a few fortunate patients. Any technological advance that reduces the required dose could bring immediate benefit, as a tenfold reduction in dose might also lower the cost tenfold. Our foam formulation for gene therapy vectors enhances gene transfer and ensures that the drug stays in place, so the same therapeutic effect could be achieved with only a fraction of the vector dose. Thus, our gene therapy platform is built for wide-scale adoption and expanded patient access.

3. **The problem of safety:** The inability to precisely specify the in vivo fate of a therapeutic gene cargo diminishes potency, challenges safety, and ultimately lowers the treatment efficacy. Gene therapy can pose significant risks, including potential oncogenesis and immune system-mediated toxicity against the vector, which can be fatal[29,30]. Foam could improve the safety profile of gene therapy, as it retains the drug at the intended target tissue, thus minimizing off-target events. Furthermore, as foam isolates the gene therapy vector in liquid-filled lamellae until delivery, the vector is shielded from recognition by the host immune system.

We chose a blend of methylcellulose and xanthan gum as our lead formulation candidate for gene therapy foam based on our in vitro screening. Both substances are already manufactured in bulk at pharmaceutical grade, which eliminates the need to develop clinical-scale manufacturing protocols, thus ensuring swift clinical translation. Our group is currently testing additional materials with foaming properties, including guar gum, for their ability to boost gene transfer of embedded gene therapy vector, so future work by us and other research teams will likely further improve the gene therapy foam formulation we are reporting here. The clinical success of this approach might also benefit from using pure oxygen instead of room air (containing ~21% oxygen) to generate the gene therapy foam. Release of additional oxygen gas from foam could further boost transgene expression in hypoxic tissue, such as solid tumor. Low oxygen supply is known to decrease the rate of intracellular mRNA-to-protein translation due to decreased ATP[31], which greatly hinders conventional gene therapy.

In summary, our findings establish that liquid foam is a highly versatile delivery platform to enhance localized gene therapy. Incorporated into the clinical workflow, this platform could shift the paradigm on how topical gene therapy is applied for the treatment of a wide range of diseases.

## Methods

### Ethical statement

Our research complies with all relevant ethical regulations: The care and use of mice in this study were approved by the Institutional Animal Care & Use Committee (IACUC) at the Fred Hutchinson Cancer Center and complied with all relevant ethical regulations for animal testing and research (Assurance #A3226–01, IACUC Protocol Number 50782).

### Cell lines

HeLa cells for in vitro transfection assays were obtained from ATCC (Cat# CCL-2) and maintained in Dulbecco's modified Eagle's medium (DMEM) containing 0.11 g/liter sodium pyruvate, 2 mM l-glutamine, 4.5 g/liter glucose, 10% fetal bovine serum (FBS), 100 U/ml penicillin, and 100 U/ml streptomycin in a humidified incubator at 5% (v/v) $CO_2$. Cells tested negative for mycoplasma using a DNA-based PCR test (DDC Medical).

### mRNA synthesis

The reporter gene mRNA, CleanCap® F-Luc (Cat# L-7202-5), was purchased from TriLink Biotechnologies (San Diego, CA).

### LNP preparation

LNPs were prepared using a previously described method with minor modifications[15]. Lipids were dissolved in ethanol at a molar ratio of 50:10:38.5:1.5 (SM-102: DSPC: cholesterol: DMG-PEG2000). mRNA was diluted with 6.25 mM sodium acetate buffer (pH 5) to 0.1 mg/mL. mRNA and lipids were combined in a Dolomite micromixer chip at a volume ratio of 3:1 (aqueous:ethanol) and flow rates of 4.5 mL/min (aqueous) and 1.5 mL/min (ethanol). The buffer of the resulting formulation was exchanged for PBS using Amicon Ultra centrifugal filters (100 K NMWL). To prepare DiD'-labeled LNPs, DiD' (1,1'-Dioctadecyl-3,3,3',3'-Tetramethylindodicarbocyanine, 4-Chlorobenzenesulfonate Salt; ThermoFisher Scientific, Cat#: D7757) was incorporated in the initial lipid mixture at a concentration of 0.05 mg/mL. Size distribution was measured via Nanoparticle Tracking Analysis using a NanoSight NS300 (Malvern), and zeta potential was determined using dynamic light scattering detected with a ZetaPALS instrument (Brookhaven). The particles were diluted 1:100 (v/v) in PBS for size measurements, and 1:33 (v/v) in water for zeta potential quantitation.

### Lentivirus production

Replication incompetent, HIV-based, VSV-G pseudotyped lentiviral particles that constitutively express firefly luciferase under the CMV promoter were purchased from BPS Biosciences (Cat# 79692). Exact titers (transduction units per ml) were provided with each shipment.

### Foam preparation

Methylcellulose HV, xanthan gum, and sodium caseinate were obtained from Modernist Pantry.

Albumin from human serum was obtained from Sigma Aldrich (Cat# A1887).

Methylcellulose, sodium caseinate, or albumin (80 mg) and xanthan gum (50 mg) were dissolved in 10 mL PBS to produce the foam precursor. To create foam, 2 mL of the foam precursor was mixed with 18 mL of air via 30 passes between two 20-mL syringes connected at 90° with a 3-way Luer adapter.

### Ex vivo transfection screening

**Horizontal transfection.** HeLa cells were plated into six-well plates and grown to 60-70% confluency. Before transfection, we removed all media from the wells and added 1 mL of fresh DMEM to each well. We then added 3 mL LNP suspension in DMEM or 3 mL foam with embedded LNPs. Two micrograms of firefly luciferase mRNA were added/well (encapsulated in LNPs). Following transfection, cells were incubated at 37 °C for 20 h. We then added 3 mL DMEM to each well, aspirated all foam and media, replaced it with 3 mL of fresh DMEM, and incubated the plates for an additional 25 h before quantitating gene expression using IVIS imaging.

**Angled transfection (see also Supplementary Fig. 2).** HeLa cells were plated into six-well plates coated with 2 mL PureCol EZ Gel collagen gel (Sigma-Aldrich) per well and grown to 60–70% confluency. Before transfection, we removed all media from wells and, with plates in a vertical orientation, added 3 mL LNP suspension in PBS or 3 mL foam with embedded LNPs. Two micrograms of firefly luciferase mRNA were added/well (encapsulated in LNPs). We then covered the plates and incubated them at 37 °C at an 120° angle for 2 h. The plates were then returned to a horizontal orientation, and we added 3 mL DMEM to each well, aspirated all foam and media, replaced it with 3 mL of fresh DMEM, and incubated the plates for an additional 25 h before quantitating gene expression using IVIS imaging.

### Foam characterization

Freshly prepared foam was characterized by Krüss USA (Matthews, NC, USA) using a Dynamic Foam Analyzer DFA100FSM (Krüss Scientific). Data were acquired and analyzed using ADVANCE software.

### Confocal microscopy

To visualize LNPs within foam lamellae (Fig. 3b, c), we freshly prepared methylcellulose/xanthan gum foam with embedded DiD'-labeled liposomes and directly added one drop of foam via an 18-gauge needle onto a 10-mm PTFE printed ring microscopy slide (Delta Microscopies, Cat#: D63417-13) before covering the slide with a 12-mm round coverslip. Fluorescent and DIC (Differential Interference Contrast) images of lamellae, which were defined as channels between two air bubbles in the foam, were collected using an Andor Dragonfly 200 High-Speed Confocal microscope (Oxford Instruments) with a ×63/1.4 (oil) objective. DiD' was excited with a 637 nm laser and collected with a 698/77 nm Cy5 bandpass emission filter. Images were acquired with an Andor Zyla 4.2 Plus sCMOS camera controlled with Fusion software (version 2.4.0.13). The resulting images were viewed and analyzed using ImageJ (Version 1.53t). For each lamella imaged, a merged image containing both the DIC and fluorescent channels was created, and brightness and contrast were adjusted (identically for all images). A region of interest (ROI) was manually drawn across the lamella between the outer edges of two bubbles using the Straight Line tool. The thickness of this ROI was set to 400 pixels, and the Plot Profile tool was used to quantify the mean gray value along its length (averaged across the 400-pixel width). These data were used to characterize the distribution of LNPs in the lamellae of the foam. Additionally, this distribution was visualized for a given lamella using the Surface Plot tool with the specified ROI to create a 3D plot of the mean gray area as it varied across the length and width of the lamella.

### TissueFAXS Imaging of intestinal samples

Mouse intestines were embedded in O.C.T. compound and snap frozen in isopentane (2-methylbutane). Following cryostat sectioning, slides were removed from −80 °C storage, placed on dry ice, and prepared for imaging by applying a drop of ProLong™ Gold Antifade reagent (Invitrogen) and a coverslip. Fluorescent and transmitted light images of the tissue sections were collected using a TissueFAXS PLUS digital pathology system/slide scanner built on a Zeiss Axio Imager Z2 upright microscope and equipped with a Hamamatsu ORCA-Flash4.0 camera. Using the TissueFAXS Imaging Software (version 7.1), ROIs along the outer edges of intestinal sections were defined for automated, high-

resolution imaging with a ×20/0.8 (air) objective. An EXFO X-Cite metal halide lamp excitation source and a Cy5 filter (excitation 590–650 nm, emission 673–762 nm) were used to detect DiD' signal. The resulting images were viewed and analyzed using ImageJ (version 1.53t). For each tissue section imaged, a merged image containing both the transmitted light and fluorescent channels was created, and brightness and contrast were adjusted (identically for all images). An ROI was manually drawn along the edge of an intestinal sample using the Segmented Line tool with a spline fit applied. The thickness of this ROI was set to 400 pixels, and the Measure tool was used to quantify the mean gray value in the fluorescent channel within the ROI. These data were then used to compare the presence of DiD' signal in intestinal samples from foam-treated and control mice.

### Animal studies
Four- to six-week-old female albino B6 (C57BL/6J-Tyr<c-2J>) mice (Strain #:000058) used in all in vivo experiments were obtained from Jackson Laboratory. For intraperitoneal (i.p.) gene transfer, mice were injected i.p. with 1.5 mL of PBS or freshly prepared foam containing an equal dose of gene therapy vector. In nonviral gene therapy studies, mice were injected with a single dose of LNPs loaded with 1 μg of luciferase-encoding mRNA. In the viral gene therapy studies, we injected a single dose of $5.25 \times 10^6$ lentiviral particles.

### Flow cytometry
Data were acquired using a BD FACSymphony™ A5 SE and analyzed with FlowJo v10.8.1. Antibodies and other staining reagents used in flow cytometry are listed in Supplementary Table 1. FACS gating strategies are shown in Supplementary Fig. 3.

### ELISA
To quantify lentiviral particles in peritoneal lavage fluid and peripheral blood (Fig. 8C), we used an HIV1 p24 ELISA kit (Abcam, Cat#: ab218268), according to manufacturer's instructions.

### Toxicity analysis
To measure potential in vivo toxicities of gene therapy foam (Fig. 6), we injected mice (10/group) i.p. with 1.5 mL of PBS or methylcellulose/xanthan gum foam. Forty-eight hours after injection, mice were anesthetized and blood was collected by retro-orbital bleed to determine the complete blood counts. Blood was also collected for serum chemistry analyses (performed by Moichor Animal Diagnostics, San Francisco, CA). Animals were then euthanized with $CO_2$ to retrieve organs, which were washed with deionized water before fixation in 4% paraformaldehyde. The tissues were processed routinely, and sections were stained with hematoxylin and eosin. The specimens were interpreted by board-certified staff pathologists, in a blinded fashion.

**In vivo bioluminescence imaging.** We used D-Luciferin (Xenogen) in PBS (15 mg/ml) as a substrate for F-luc. Bioluminescence images were collected with a Xenogen IVIS Spectrum Imaging System (Perkin Elmer). Living Image software version 4.7.3 (Perkin Elmer) was used to acquire (and later quantitate) the data 10 min after i.p. injection of D-luciferin into animals anesthetized with 150 mg/kg of 2% isoflurane (Forane, Baxter Healthcare). Acquisition times ranged from 10 sec to 5 min.

**Statistics and reproducibility.** The statistical significance of observed differences was analyzed using an unpaired, two-tailed Student's $t$ test. The $P$ values for each measurement are listed in the figures or figure legends. All statistical analyses were performed using GraphPad Prism software version 9.0. To evaluate whether foam-embedded LNPs were homogenously dispersed within the liquid lamellae (Fig. 3b–d), we calculated the deciles of the average particle density for each image ($n = 27$) by dividing the distance between two air bubbles into ten equal

groups. The boxplots of each decile were generated for visualization. To show whether there is any difference in the average LNP density between those decile groups, we conducted an overall test using the nonparametric Friedman test and pairwise comparisons using the Wilcoxon Signed rank test. No statistical method was used to predetermine sample size. No data were excluded from the analyses. To ensure reproducibility, we randomized mice before injection with a gene therapy vector (in suspension or foam). The Investigators were not blinded to allocation during experiments and outcome assessment.

### Reporting summary
Further information on research design is available in the Nature Portfolio Reporting Summary linked to this article.

## Data availability
The data generated in this study have been deposited with Figshare[32]. Source data are provided with this paper.

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

## Acknowledgements

We thank the Fred Hutch Experimental Histopathology Core staff for assistance in the cryostat sectioning and Dr. Amanda Koehne, DVM, PhD, DACVP, for her help with the toxicity analysis. This work was supported in part by the *Fred Hutch Immunotherapy Initiative* with funds provided by the Bezos Family Foundation. This research was also supported by the Experimental Histopathology Shared Resource, and the Comparative Medicine Shared Resource of the Fred Hutch/University of Washington Cancer Consortium (P30 CA015704).

## Author contributions

K.F. and S.B.S. helped conceive the study, and designed and performed the experiments. N.M. and Q.V.W. conducted the statistical analysis of LNP dispersion within the foam lamellae (Fig. 3b–d), and M.T.S. conceived the study, helped design the experiments, and wrote the manuscript.

## Competing interests

The authors declare no competing interests.
