## [Peer Review File · Nature Communications]

REVIEWER COMMENTS

Reviewer #1 (Remarks to the Author):

In my opinion, the paper „Liquid foam improves potency and safety of gene therapy vectors" presents a brilliant idea. Of course, use in therapy needs more research i.e.: optimization of concentration of vectors or composition of foam. The article presents good work, is very well introduced and elaborated. I have only one small remark. In the method section the authors entitled parts of Foam synthesis. In my opinion better will be Foam preparation (synthesis is kind of chemical reaction)

In Conclusion, I recommend the present article for publishing.

Reviewer #2 (Remarks to the Author):

Comments to the Author

The paper "Liquid foam improves potency and safety of gene therapy vectors" is written well. Despite the substantial efforts and the fact that the material provided is fresh and perfectly suited to the journal's scope, "minor revision" is still needed to further raise the manuscript's level of scientific excellence. The following recommendations must be addressed.

1. Please check the misspellings in the whole text. Check all text Grammarly.
2. Author must check the abstract part; some sentences are poorly written (Reframe).
3. Reframe the introduction part for better clarity.
4. Author must cite the following article: <https://doi.org/10.1208/s12249-022-02390-x> : <https://doi.org/10.1016/j.jddst.2022.103533>.
5. In place of mouse study author should write animal study.
6. How does the efficacy of gene delivery using foam-based vectors compare to traditional liquid-based vectors in various cell types or tissues?
7. The lipid nanoparticles evaluation is missing and what was the foam bubble size. Please explain ?

Reviewer #3 (Remarks to the Author):

The strengths of the manuscript are that the authors demonstrate a clear advantage in transfection efficiency for lipid nanoparticles embedded in methylcellulose foam over liquid suspension, characterization data of the foam formulation to support their proposed mechanism for its enhanced efficiency, promising initial in vivo data based on intraperitoneal delivery, and versatility considering the demonstration of the technology with LNPs and viral vectors. The weaknesses of the article include a lack of cellular level data comparing delivery efficiency and use of a skin delivery model (dermabrasion) which is limited in its translational relevance.

Major comments:

- The authors present promising in vivo data using in vivo imaging system and fluorescent microscopy. The study would benefit from adding cellular level delivery data by using flow cytometry to analyze fluorescence in different cell types from dissociated tissues to show whether cellular delivery is increased using the foam technology.

- There are significant concerns with the skin delivery model. Dermabrasion is rarely used in clinical practice and removes multiple nucleated layers of the skin. In thin mouse skin, it is possible that dermabrasion would ablate almost the entire epidermis which is quite thin. The authors should show the histology of skin post-dermabrasion to prove that a full thickness wound has not been created. Dermabrasion would not be clinically practical for a therapeutic approach in humans given invasiveness, pain, and potential for adverse events. Thus, it is not realistic to perform dermabrasion to deliver a gene product for most genetic skin conditions. It would be useful to look at the performance of the foam technology at delivering therapeutics to intact skin. For example, since the foam technology would potentially allow the therapeutic product to better adhere to the skin of a moving mouse, might that allow for improved absorption and better delivery to epidermal stem cells at the basal layer of the epidermis? The authors should show using immunofluorescence which cell layers take up cargo.

- It is hard to interpret the pathological findings shown in Fig 7B-C and described as mild or minor by the authors without having a direct comparison to LNP in liquid suspension (as in Figure 9c). Having the direct comparison would allow the authors to better support their claim of enhanced delivery efficiency with either comparable or enhanced safety.

Minor comments:

- Figure 4 d needs a clearer explanation of the deciles, and the results should more clearly explain the functional implication of this variation in nanoparticle density as a function of spatial location

- For the “angled transfection” subheading of the Methods section, it may be helpful to have a supplementary figure illustrating the method, because it is a little bit difficult to understand the sequence of experimental steps based on the methods section.

- The authors may consider moving Figure 2 to be the last figure describing the potential clinical translation because it corresponds more with the discussion section of a manuscript than the introduction.

Reviewer #1 (Remarks to the Author):

In my opinion, the paper „Liquid foam improves potency and safety of gene therapy vectors" presents a brilliant idea. Of course, use in therapy needs more research i.e.: optimization of concentration of vectors or composition of foam. The article presents good work, is very well introduced and elaborated. I have only one small remark. In the method section the authors entitled parts of Foam synthesis. In my opinion better will be Foam preparation (synthesis is kind of chemical reaction).

In Conclusion, I recommend the present article for publishing.

We appreciate these encouraging words and would like to thank the Reviewer for their time. We revised the Methods section according to the Reviewer's suggestion.

Reviewer #2 (Remarks to the Author):

The paper "Liquid foam improves potency and safety of gene therapy vectors" is written well. Despite the substantial efforts and the fact that the material provided is fresh and perfectly suited to the journal's scope, "minor revision" is still needed to further raise the manuscript's level of scientific excellence. The following recommendations must be addressed.

1. Please check the misspellings in the whole text. Check all text Grammarly.

The revised manuscript was edited and proofread by a professional editing service and should not contain any spelling mistakes or grammatical errors. Also, the entire manuscript was spell-checked in MS Word. Please note that we used American spelling throughout, not British spelling.

2. Author must check the abstract part; some sentences are poorly written (Reframe).

We re-wrote sections of the abstract to improve readability.

3. Reframe the introduction part for better clarity.

The introduction was revised for clarity, conciseness, and coherence. Also, as suggested by Reviewer 3, we moved the envisioned clinical workflow (originally **Fig. 2**) to the Discussion section (now Fig. 9 of the revised manuscript).

4. Author must cite the following article: <https://doi.org/10.1208/s12249-022-02390-x> : <https://doi.org/10.1016/j.jddst.2022.103533>.

This reference was added to the revised manuscript (after the first sentence: "Foam is becoming a prominent delivery system for small molecule drugs to treat various medical conditions").

5. In place of mouse study author should write animal study.

We revised this accordingly.

6. How does the efficacy of gene delivery using foam-based vectors compare to traditional liquid-based vectors in various cell types or tissues?

We included a direct side-by-side comparison between suspension- and foam-delivered gene therapy in all our studies. The key in vivo experiments are summarized below. In the "suspension" treatment arms, vector was resuspended in sterile Phosphate Buffered Saline

(PBS) – similar to the current clinical gene therapy protocols. In the “foam” treatment groups, an equal dose of vector was embedded in methylcellulose/xanthan gum foam. We found that foam injections resulted in an overall average 3.2-fold higher transfection rate compared to liquid-based gene therapy, when using a nonviral gene therapy vector (**Fig. 1** below). Foam amplified the transfection rate of viral gene therapy vector by an average 10.2-fold when compared to liquid suspension (**Fig. 2** below).

Fig. 1: Foam as a carrier system for gene therapy vectors can improve in situ gene transfer. Immunocompetent C57BL/6 albino mice were injected intraperitoneally with a single dose of LNPs suspended in PBS or incorporated into methylcellulose foam. To noninvasively track gene expression in vivo, LNPs were loaded with luciferase-encoding mRNA. **(a)** Representative sequential bioluminescence imaging of gene expression. **(b)** Box plots showing photon counts from luciferase activity. The boxes represent the mean values and the line in the box represents the median. The bars across the boxes show the minimum and maximum values. Whiskers represent 95% confidence intervals. N = 10 biologically independent samples. P < 0.05 was considered significant (*).

Fig. 2: Foam amplifies lentivirus-mediated gene transfer while limiting undesirable systemic exposure. C57BL/6 albino mice were injected intraperitoneally with a single dose of 5.25×10^6 lentiviral particles (LV) suspended in PBS or incorporated into methylcellulose foam. To noninvasively track gene expression in vivo, LV expressed luciferase. (a) Representative sequential bioluminescence imaging of gene expression. (b) Box plots showing photon counts from luciferase activity. N = 10 biologically independent samples. P < 0.05 was considered significant (*). (c) Lentivirus vector concentrations in the peritoneal fluid and peripheral blood 3 hours post injections. N = 10 biologically independent samples. P < 0.05 was considered significant (*).

7. The lipid nanoparticles evaluation is missing and what was the foam bubble size. Please explain ?

We are now adding a figure summarizing the physicochemical characterization of lipid nanoparticles (Supplementary Fig. 1, see **Fig. 3** below) to our revised manuscript. Based on three independently manufactured LNP batches, the average particle size is 92.3 nm, the average zeta potential is 4.5 mV and we achieve 98% mRNA encapsulation. The LNP formulation used in our studies is based on the formulation of Moderna's COVID-19 mRNA vaccine (Sabnis S et al., *Mol. Ther.* 26, 2018) and has therefore been studied extensively.

The mean bubble size immediately after synthesis is mentioned in our manuscript: $10,528 \pm 455 \mu\text{m}^2$.

- Size Distribution (Nanosight): 92.3 ± 1.6 nm
- Zeta Potential: 4.5 ± 2.2 mV
- MRNA Encapsulation: 98-99%

Fig. 3: Physicochemical characterization of Lipid nanoparticles (LNPs). Size distribution was measured via Nanoparticle Tracking Analysis using a NanoSight NS300 (Malvern) and zeta potential was determined using dynamic light scattering detected with a ZetaPALS instrument (Brookhaven). The particles were diluted 1:100 (v/v) in PBS for size measurements, and 1:33 (v/v) in water for zeta potential quantitation. Size distribution was determined for three independently prepared batches of LNPs. For each batch, three consecutive videos (60 s each) were captured under constant flow conditions. MRNA encapsulation efficiency was determined using a Qubit RNA High Sensitivity Assay Kit (Invitrogen). LNPs were incubated with the Qubit RNA reagent in the presence and absence of 1% Triton X-100. Fluorescence intensities for total mRNA after release from LNPs by Triton X-100 were compared to fluorescence intensities for unencapsulated mRNA measured in the absence of Triton X-100.

We appreciate the constructive criticisms, and hope that our responses appropriately address the issues the reviewer raised. The changes made in response to them have substantially improved our manuscript.

Reviewer #3 (Remarks to the Author):

The strengths of the manuscript are that the authors demonstrate a clear advantage in transfection efficiency for lipid nanoparticles embedded in methylcellulose foam over liquid suspension, characterization data of the foam formulation to support their proposed mechanism for its enhanced efficiency, promising initial in vivo data based on intraperitoneal delivery, and versatility considering the demonstration of the technology with LNPs and viral vectors. The weaknesses of the article include a lack of cellular level data comparing delivery efficiency and use of a skin delivery model (dermabrasion) which is limited in its translational relevance.

Major comments:

- The authors present promising in vivo data using in vivo imaging system and fluorescent microscopy. The study would benefit from adding cellular level delivery data by using flow cytometry to analyze fluorescence in different cell types from dissociated tissues to show whether cellular delivery is increased using the foam technology.

We now include a flow cytometry-based analysis to test differences in transgene expression following vector delivery in suspension versus foam at a cellular level (Fig. 1 below, new **Supplementary Fig. 3**. Twenty-four hours after a bolus i.p. injection of mCherry mRNA LNPs in PBS or methylcellulose foam, peritoneal cavity cells or splenocytes were isolated from mice. To recover peritoneal cells, we used lavage according to an established protocol:

<https://www.ncbi.nlm.nih.gov/pmc/articles/PMC3152216/>

Single-cell suspensions were stained with antibodies against CD45, MHCII, and F4/80 (phagocyte panel) and differences in mCherry expression levels between treatment groups were quantitated by flow cytometry. We found that the use of foam as carrier substantially improved gene transfer into phagocytes (2.94-fold increase in peritoneal phagocytes, 3-fold increase in spleen macrophages.)

Fig. 1: Flow cytometric quantification of in vivo transfection rates into peritoneal and splenic macrophages using foam versus suspension. Immunocompetent C57BL/6 albino mice were injected intraperitoneally with a single dose of LNPs suspended in PBS or incorporated into methylcellulose foam. To track gene expression by flow cytometry, LNPs were loaded with mRNA encoding mCherry (10 μ g/mouse). Twenty-four hours after injection, peritoneal exudate cells and splenocytes were harvested. Flow cytometric quantitation of in vivo transfection rates in macrophages are summarized in (a) and (b), respectively. Gating strategies for flow cytometric analysis are shown on top of each panel. N = 3 biologically independent samples. P < 0.05 was considered significant (*).

- There are significant concerns with the skin delivery model. Dermabrasion is rarely used in clinical practice and removes multiple nucleated layers of the skin. In thin mouse skin, it is possible that dermabrasion would ablate almost the entire epidermis which is quite thin. The authors should show the histology of skin post-dermabrasion to prove that a full thickness wound has not been created. Dermabrasion would not be clinically practical for a therapeutic approach in humans given invasiveness, pain, and potential for adverse events. Thus, it is not realistic to perform dermabrasion to deliver a gene product for most genetic skin conditions. It would be useful to look at the performance of the foam technology at delivering therapeutics to intact skin.

Based on the reviewer's suggestion, we asked our histopathology team at Fred Hutchinson Cancer Center to prepare H&E-stained sections of mouse skin with or without dermabrasion. A board-certified pathologist, A. Koehne, DVM, PhD, DACVP, conducted the pathological evaluation in a blinded fashion. Her results are summarized in Fig. 2 below (new **Supplementary Fig. 5**). In summary, the dermabrasion procedure (using a sterile surgical blade to scrape the surface of the skin) efficiently removed the stratum corneum. In some sections the procedure caused an ulceration that extended into the superficial dermis. As explained in more detail below, in all our envisioned clinical applications for gene therapy foam for skin conditions, patients will present with some degree of damaged skin, which we tried to recapitulate in our mouse studies using dermabrasion.

Fig. 2 Histopathological evaluation of mouse skin before and after dermabrasion (all mice used for skin gene therapy experiments in **Fig. 7** underwent mechanical dermabrasion). A square (~1 cm diameter) of the dorsal side of the skin was shaved and the surface of the skin was scraped with a sterile surgical blade (or left untouched). Mice were then immediately euthanized and the skin was excised and processed for sectioning and hematoxylin and eosin (H&E) staining. A board-certified comparative pathologist at Fred Hutch Cancer Center (A. Koehne, DVM, PhD, DACVP) then performed the histopathological evaluation. In summary, the dermabrasion procedure efficiently removed the stratum corneum. In some sections the procedure caused an ulceration that extended into the superficial dermis. Shown are three representative examples for each group. Scale bar: 100 μ m.

Following the reviewer’s request, we also repeated our skin gene therapy studies on intact mouse skin. As shown in Fig. 3 below (new **Supplementary Fig. 4** of the revised manuscript), we could not achieve gene transfer above background levels when delivering mRNA lipid nanoparticles in foam without dermabrasion.

This observation is agreement with the current literature (recently summarized in the Review Article: “*Gene Delivery to the Skin – How Far Have We Come?*”)¹. The absorption of biomacromolecules into intact skin is highly restricted (molecular weight ≤ 800 Da and moderately lipophilic). This, however, has not stopped development of cutaneous gene therapy

for patients as the clinical need is treatment of diseased (damaged or blistered) skin. Below is a summary of ongoing clinical trials for various skin conditions by gene therapy companies.

Key companies and their lead assets					
Company	Drug	Phase	Technology	Indication	Designation
Krystal Biotech	Beremagene geperpavec (B-VEC)	Phase III	Skin TARgeted Delivery (STAR-D) platform	Dystrophic Epidermolysis Bullosa	Orphan Drug Designation Fast Track Designation Rare Pediatric Designation Regenerative Medicine Advanced Therapy (RMAT) PRiority Medicines (PRIME)
Castle Creek Biosciences	D-Fi	Phase III	Autologous fibroblasts technology	Dystrophic Epidermolysis Bullosa	Orphan Drug Designation Rare Pediatric Disease Designation Fast Track Designation Regenerative Medicine Advanced Therapy (RMAT)
Castle Creek Biosciences	FCX-013	Phase I/II	Autologous fibroblasts technology	Localized Scleroderma	Orphan Drug Designation Rare Pediatric Disease Designation Fast Track Designation
Abeona Therapeutics	EB-101	Phase III	Gene Transfer	Recessive Dystrophic Epidermolysis Bullosa	Regenerative Medicine Advanced Therapy Breakthrough Therapy Rare Pediatric Disease Designations Orphan Drug Designation
Amryt Pharma	AP103	N/A	Gene therapy platform	Dystrophic Epidermolysis Bullosa	Orphan Drug Designation
Avita Medical	N/A	N/A	RECELL System technology	N/A	N/A
Aziltra	ATR-12	N/A	N/A	Netherton Syndrome	Rare Pediatric Disease Designation

@Delveinsight.com

In all the mentioned indications, skin lesions are treated with gene therapy to promote wound healing. One prime example is Vyjuvek®, which was recently FDA-approved as the first topical gene therapy for treatment of wounds in patients with dystrophic epidermolysis bullosa.

We believe that, based on our proof-of-concept in vivo data, gene therapy foam could be developed for various skin diseases, including epidermolysis bullosa (delivering the COL7A1 gene), autosomal recessive congenital ichthyosis (by overexpressing the TGM1 gene), or Netherton disease (by expressing the SPINK5 gene). In addition, we hope to partner with industry to develop gene therapy foam to accelerate wound healing in diabetics, which are very slow to heal or do not heal at all.

Netherton syndrome

Therefore, in all our envisioned clinical applications for gene therapy foam for skin conditions, patients will present with some degree of damaged skin, which we tried to recapitulate in our mouse studies using dermabrasion.

- It is hard to interpret the pathological findings shown in Fig 7B-C and described as mild or minor by the authors without having a direct comparison to LNP in liquid suspension (as in Figure 9c). Having the direct comparison would allow the authors to better support their claim of enhanced delivery efficiency with either comparable or enhanced safety.

Based on the reviewer's comment, we realized that we did not make it clear in the figure legend that the toxicity study shown in Fig 7 of the original manuscript measured the biocompatibility of methylcellulose-based foam. Thus, no mRNA LNPs were added to the foam in any treatment group. We are certain that the LNP formulation we chose is safe as we based it on the formulation of Moderna's COVID-19 mRNA vaccine. We have also shown in our previous studies that the various mRNA nanoparticle formulations we injected into the peritoneal cavity are biocompatible^{2,3}.

Obviously, the overall safety of gene therapy foam will need to be confirmed individually for each therapeutic gene therapy vector (nonviral or viral) once this technology reaches IND-enabling studies in large animal models, which is not the focus of the current study. Our goal is to introduce foam as a new platform for topical gene therapy and confirm that the foam formulation (i.e. the carrier) we chose as our lead candidate is safe (independent of the potential gene therapy cargo).

We revised our figure legend accordingly to:

Methylcellulose-based foam is biocompatible. These panels summarize a pathology report prepared by a Comparative Pathologist at Fred Hutch. Two days after a bolus injection of methylcellulose foam or PBS into the peritoneal cavity of mice, various key organs close to the peritoneum as well as blood were isolated for a double-blinded analysis. **(a)** Serum chemistry and blood counts. Data are mean \pm s.d.; N = 8 biologically independent samples. **(b)** Representative hematoxylin and eosin (H&E)-stained sections of the mesenteric fat from mice (4/10) that exhibited mild regional infiltrates of neutrophils and histiocytes in response to foam injection. **(c)** Representative image of the marginal zone of a spleen 48 hours after intraperitoneal foam injection. 40 \times objective.

Minor comments:

- Figure 4 d needs a clearer explanation of the deciles, and the results should more clearly explain the functional implication of this variation in nanoparticle density as a function of spatial location

We forwarded this comment to our lead biostatisticians (and co-authors), Dr. Qian Wu and Dr. Ningxin Ma, who did the calculations for the original figure. Below are their comments:

“We thank the reviewers for their suggestions to better explain what we did for Figure 4D. We revised the legend by adding more details. We would like to clarify that Figure 4D evaluates whether the nanoparticle density was equally distributed or not. Since our goal is not to state the variation is a function of spatial location, we removed the kernel smoothing curve (blue curve) to avoid the confusion.

For each lamella, we equally divided the distance between two bubbles into ten segments, and calculated the average fluorescence intensity within each segment. Each box-plot combines 27 independent lamellae together (i.e., there are 27 data points for each segment) and compares the mean density per segment across 10 segments from left to right (In Figure 4D, distance decile = 1 refers to the first segment from the left side, distance decile = 5 and 6 refer to the middle segments, and distance decile = 10 refers to the last segment from the left side, which is also the first segment from the right side). We observed a statistically significant difference between the 10 segments by using an overall nonparametric Friedman test, giving a p -value <0.0001 .

We changed the legend of Figure 4 (now Figure 3 in the revised manuscript) to:

For each image, we divided the distance between two air bubbles into ten equal segments and calculated the average particle density within each segment. The boxplots show the mean fluorescence intensities of each segment for 27 independent images (distance decile 1 refers to the first segment from the left side and distance decile 10 refers to the 10th segment from the left side, which is also the first segment from the right side).”

- For the “angled transfection” subheading of the Methods section, it may be helpful to have a supplementary figure illustrating the method, because it is a little bit difficult to understand the sequence of experimental steps based on the methods section.

We are now illustrating each step of the “angled transfection” in the newly added **Supplementary Fig. 2** (see Fig. 5 below).

- The authors may consider moving Figure 2 to be the last figure describing the potential clinical translation because it corresponds more with the discussion section of a manuscript than the introduction.

We agree and moved this illustration to the discussion section in our revised manuscript.

We appreciate the constructive criticisms, and hope that our responses appropriately address the issues the reviewer raised.

References

1. Ain, Q.U., Campos, E.V.R., Huynh, A., Witzigmann, D. & Hedtrich, S. Gene Delivery to the Skin - How Far Have We Come? *Trends Biotechnol* **39**, 474-487 (2021).
2. Hao, S., *et al.* BiTE secretion from in situ-programmed myeloid cells results in tumor-retained pharmacology. *J Control Release* **342**, 14-25 (2022).
3. Parayath, N.N., *et al.* Genetic in situ engineering of myeloid regulatory cells controls inflammation in autoimmunity. *J Control Release* **339**, 553-561 (2021).

REVIEWER COMMENTS

Reviewer #2 (Remarks to the Author):

All the said changes are incorporated in the revised manuscript file. The manuscript is suitable for publication.

Reviewer #3 (Remarks to the Author):

The authors provide data to show that the gel formulation does not deliver to intact skin.

The references to therapies for EB are entirely focused on a disease which features ulceration equivalent to that achieved with destruction of mouse skin with dermabrasion.

The pathology images shown are of low quality. All examples show evidence of full thickness skin damage with ulceration. While it is a true statement that dermabrasion "successfully removed the stratum corneum" it also mis-characterizes what is shown. A skin pathologist could further assist. Example 1 shows mid-epidermal lysis which is the level throughout most of the samples shown. The multiple ulcerations in every sample are likely needed for delivery.

It is inappropriate to say that dermabrasion of thin mouse skin is equivalent to the same approach in thicker human skin. The more accurate title for figure 3 should be "Gene transfer into the skin depends upon full thickness wounding". If the authors wish to include skin data, the lack of efficacy in normal skin should be in the manuscript, rather than the supplement.

Minor comments:

The flow cytometry data in Fig S3 needs to show an untreated control animal as a negative control for mCherry fluorescence.

The image resolution of the annotated H&E images in Fig S5 needs to be better to be able to see the features and read the labels.

Reviewer #2 (Remarks to the Author):

All the said changes are incorporated in the revised manuscript file. The manuscript is suitable for publication.

Reviewer #3 (Remarks to the Author):

The authors provide data to show that the gel formulation does not deliver to intact skin. The references to therapies for EB are entirely focused on a disease which features ulceration equivalent to that achieved with destruction of mouse skin with dermabrasion.

The pathology images shown are of low quality. All examples show evidence of full thickness skin damage with ulceration. While it is a true statement that dermabrasion "successfully removed the stratum corneum" it also mis-characterizes what is shown. A skin pathologist could further assist. Example 1 shows mid-epidermal lysis which is the level throughout most of the samples shown. The multiple ulcerations in every sample are likely needed for delivery.

It is inappropriate to say that dermabrasion of thin mouse skin is equivalent to the same approach in thicker human skin. The more accurate title for figure 3 should be "Gene transfer into the skin depends upon full thickness wounding". If the authors wish to include skin data, the lack of efficacy in normal skin should be in the manuscript, rather than the supplement.

Following the Editorial Board's recommendation, we removed the skin application from the revised manuscript and focus only on intraperitoneal injection.

Minor comments:

The flow cytometry data in Fig S3 needs to show an untreated control animal as a negative control for mCherry fluorescence.

We are now showing this control group in the revised Fig. S3.